# Endogenously Emerging Gender Pay Gap in an Experimental Teamwork Setting

**Özgür Gürerk [1],\*** 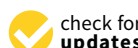**, Bernd Irlenbusch [2] and Bettina Rockenbach [3]**

[1]    School of Business and Economics, RWTH Aachen University, 52056 Aachen, Germany
[2]    Department of Management and C-SEB, University of Cologne, 50923 Cologne, Germany;
        bernd.irlenbusch@uni-koeln.de
[3]    Department of Economics and C-SEB, University of Cologne, 50923 Cologne, Germany;
        bettina.rockenbach@uni-koeln.de
\*    Correspondence: ozgur.gurerk@rwth-aachen.de; Tel.: +49-241-8096-702

**Abstract:** We studied gender diversity and performance in endogenously formed teams in a repeated teamwork setting. In our experiment, the participants ($N$ = 168, 84 women and 84 men) chose whether to perform a cooperative task only with members of the own gender or in a mixed-gender team. We found that independent of the choice of team, in the initial period, men contributed significantly more to the team projects than women. Men preferred the successful men-only teams in the subsequent periods, resulting in significantly higher profits for men compared to women. This endogenously emerging "gender pay gap" only closed over time.

**Keywords:** gender differences; endogenously formed teams; cooperation; punishment

## 1. Introduction

"Gender equality is a moral and a business imperative." This succinct statement opening the blurb of Bohnet's [1] recent book summarizes a widely shared view, not only in the business world but also in most western societies. Yet, the appropriate means to achieve gender equality in the workplace are not at all trivial and have many facets. Bohnet's book presents an impressive collection of different (institutional) design features, based on behavioral insights that help to overcome biases and stereotypes. In addition to these (in part soft) interventions to promote gender equality, many countries have embarked on more radical and schematic interventions by implementing quotas for female representation in management boards and committees.[1] The implementation of quotas addresses the moral facet of gender equality, independent of its (direct) economic consequences.

The economic consequences of gender diversity of work teams have been studied in the field as well as in experimental laboratory contexts. Ahern and Dittmar [3] used data from a natural field experiment to investigate the implementation of gender quotas in Norway. They reported a decline in stock prices and operating profits after the implementation of the gender quota in boards and attributed this to lesser CEO experience of the newly appointed female directors. In a field experiment, Hoogendoorn et al. [4] reported that undergraduate business teams with an equal gender mix perform better than male-dominated teams concerning sales and profits. The authors attributed this effect to more intensive monitoring and more equalized learning in mixed teams than in homogeneous teams.

---

[1]    As reported in a New York Times article [2], in 2015, Germany passed a law that requires the 100 plus biggest companies of the country to give at least 30 percent of the seats in their corporate boards to women.

Azmat and Petrongolo [5] present an overview of laboratory as well as field experiments on the economic consequences of gender diversity in work teams. While they report a number of relevant findings, they point to the important research issue that team compositions in the workplace generally evolve endogenously[2]: "One of the main problems with studying gender and groups is that groups are typically formed in an endogenous way. While experiments can go some way to solve this issue through random assignment into groups, they tend to create an artificial environment, in which it becomes difficult to distinguish group diversity and group dynamics." (p. 38).

In this paper, we address this issue by asking whether and how gender diversity affects team performance in endogenously formed teams. In a laboratory experiment, male and female participants chose whether to perform a task in a team with members of the same (own) gender (women-only and men-only) or whether to perform the task in a mixed-gender team. Repeated choice allowed us to study the dynamics over time. The experimental task asked for individually costly contributions to a team project that benefited all team members equally, independent of the individual contributions. Our experimental setup allowed investigation of the preferences of women and men for same- or mixed-gender teams, in their initial choice as well as over time. Moreover, we studied whether and how the gender composition of a team influences women's and men's contributions and payoffs and investigated whether endogenously formed same-gender teams perform differently to endogenously formed mixed-gender teams.

Remarkably, we found an endogenously occurring "gender pay gap", primarily driven by gender differences in the initial behavior. In the first round, men exhibited a clear preference for the mixed-gender team, while women seemed to be indifferent to the mixed-gender and the women-only team. Independent of the team choice, men contributed significantly more to the team project than women. These initial differences seem to induce men to predominantly choose the more successful men-only team in the further rounds, resulting in significantly higher profits for men compared to women. It took until the second half of the thirty repetitions for women to "recover" from the disadvantage of the low initial contributions and for the "gender pay gap" to close.

The article is organized as follows. In the next section, we describe the experimental game. In Section 3, we review the main theoretical benchmarks and derive our hypotheses. Section 4 provides the specific experimental design and the experimental procedures. Section 5 contains the results and finally, in Section 6, we summarize and make our conclusions.

## 2. The Experimental Game

In our experiment groups of twelve players, six females and six males, repeatedly interacted over 30 periods in a partner matching. The groups constituted the independent experimental observation. The experiment proceeded in three stages.

### 2.1. Stage 1: Team Choice Stage

In the team choice stage, each player decides on the team membership. Women could join either the women-only team, abbreviated as the W-team or the mixed WM-team, men could join either the men-only M-team or the mixed WM-team. Thus, in any given period, a maximum of 12 players could join the WM-Team, while the maximum possible number of players, in the W-team and the M-team, was six. After players made their team choice, and before the contribution stage started, each player learnt how many individuals were in each team. Players also learnt how many men and women joined the WM-team.

---

[2]　Endogenously formed teams have rarely been studied with regard to gender diversity. In this respect, we are only aware of an attitude survey conducted by Chatman and O'Reilly [6] who report that women express a greater likelihood of leaving homogeneous groups than do men in the clothing and retail industries.

## 2.2. Stage 2: Contribution Stage

In the contribution stage, each player receives an endowment of 20 tokens. If only one player selected a team, no team game was played, and the single player's endowment was added automatically to their private account. If a team had two or more members, then a team game was played within the team, independently of the other teams in the group. The team game simulates team production with conflicting individual and collective interests through a public goods game. Team members decide how many of their 20 tokens to invest into the team account. Each token invested creates a benefit of 1.6 for the entire team, which is then equally distributed among the team members. Thus, investing in the team account benefits the entire team. Yet, the individual return (marginal per capita return MPCR) on the invested token is lower than 1 (MPCR = 1.6 divided by the number of team members). This means that for each player it is individually more profitable to save the token in the private account (with a private return of 1 token), rather than putting it into the team account (with a private return of less than 1 token). From the collective perspective of the team, however, contributions increase efficiency. The discrepancy in the benefits of investment between the team and the individual constitute the conflict in individual and collective interests.

We decided to hold the productivity (return) of 1.6 constant for each contributed token, independent of the number of members in a team. This design choice eliminated the possible motivation of joining a larger team for pure efficiency reasons. Therefore, the MPCR in our game varies with the number of players in a team. For a player, who is in a team with only one other player, the MPCR is 0.8 (= 1.6/2), while it decreases linearly as the team grows. In a team with 12 members, this amounts to 0.13 (= 1.6/12).[3]

## 2.3. Stage 3: Punishment Stage

In the punishment stage, each player receives an additional 20 tokens. If there is only one player in a team, there is no action taken for this player. If there are two players or more, each player learns the individual contributions of all other members of the team and their genders. This data was provided without any subject IDs and ordered randomly, so that it was not possible to track an individual over periods. Each player can allocate the 20 tokens at their discretion to punish others in the team or save the tokens in their private account. The sum of allocated tokens in one period could not exceed 20 tokens. We apply a commonly used experimental punishment technology with 1:3 effectiveness. This means, each allocated punishment token cost the punishing player one token, and reduced the payoff of the punished player by three tokens. Each player can allocate punishment tokens to one or several other team members. In particular, it is possible for a player to get punished by several team members. To reduce the risk of bankruptcy, we provided each player with an additional endowment of 400 tokens at the start of the experiment.

At the end of stage 3, players were informed about other players' individual contributions, received punishment tokens, and payoffs; both in their own team and the other team. From period 2 on, prior to stage 1, players were provided with a table displaying the average numbers (contributions, punishment and payoff) of all three teams in the previous period(s).

## 3. Theoretical Considerations and Hypotheses

### 3.1. Nash Equilibrium with Myopic and Selfish Preferences

If we consider the stage game, assuming that each player is a money maximizer with myopic, selfish preferences, then no player has an incentive to punish any other player since punishment is

---

[3] Some previous studies share the basic design with the current study, in particular with respect to varying the MPCR (see, e.g., Gürerk et al. [7,8]). In these studies, one observes large teams emerging, which means that MPCRs endogenously emerge to be small.

costly. This implies that no player contributes to the public good in the first stage. Since the structure is the same in all teams, a player is indifferent to choosing between the only own-gender team or the mixed-gender team. The payoff of a player then equals to 40 tokens. Since the number of periods is finite, and the players know this, by backward induction the predicted behavior of the last period unravels to all periods.[4]

### 3.2. Socially Optimal Behavior

If one assumes that there is more than one player in a team, payoffs would be maximized when all players contribute fully to the public good, and no player punishes any other player. In this case, the payoff of each player amounts to 52 tokens, regardless of the number of players in the team.

### 3.3. Hypotheses on Gender Differences

The rapidly growing literature on gender-related characteristics in economic environments reveals many cases with inconclusive gender differences (see Croson and Gneezy [9] for an excellent overview, and Balliet et al. [10] for a large meta-study on cooperation and gender in exogenously formed groups). One of the robust findings seems to be that women are more responsive to the experimental conditions, in particular, when they know the gender of their counterparts. Additionally, there is evidence that both genders prefer to interact with women. In a two-person winner-takes-it-all competition, for example, Geraldes [11] showed that 65% of men and 80% of women choose to be paired with a woman. The preference to be paired with a woman does not depend on the deciding subject's "confidence-level" on the task, for either gender. These insights lead to Hypothesis 1.

**Hypothesis 1.** *a. Women tend to prefer the W-team to the WM-team. b. Men tend to prefer the WM-team to the M-team.*

Dufwenberg and Muren [12] studied the influence of gender composition in groups of three dictators allocating money between them and a fourth person. They found that groups are more generous and egalitarian when the dictator group has a majority of women. In a business game, Apesteguia et al. [13] studied how the gender composition of teams affects their economic performance. While they found that teams formed by three women are significantly outperformed by any other gender combination, they attributed this to women teams being less aggressive and investing more in social sustainability initiatives. Chatman and O'Reilly [6] report that women are more cooperative in all-female workgroups. Together with the finding that women seem to be more responsive to the experimental conditions, especially to knowing the gender of their counterparts (Croson and Gneezy [9]), we formed Hypothesis 2a.

Charness and Rustichini [14] report that in a two-person prisoners' dilemma game with an audience, men are more cooperative when (passive) women are watching than when (passive) men are watching. This finding underlies Hypothesis 2b.

**Hypothesis 2.** *a. Women contribute more in the W-team than in the WM-team. b. Men contribute more in the WM-team than in the M-team.*

Nowell and Tinkler [15] reported that exogenously formed, all-female groups are more cooperative than either exogenously formed all-male or exogenously formed mixed-gender groups. There is, however, also evidence pointing in the opposite direction (e.g., Brown-Kruse and Hummels [16]) or

---

[4] From previous experimental studies, we know that even in finitely repeated games players show a behavior that is more in line with predictions from an infinitely repeated game. Thus, the threats of no cooperation in the future, punishment, or leaving the team might lead to more cooperative behavior. There are, however, no apparent differences for our three teams concerning these patterns of behavior.

reporting no difference at all (Mason et al. [17]). Due to these inconclusive findings, we abstained from formulating a hypothesis on which team (women-only, men-only, or mixed) is more cooperative and treated this as an explorative question.

## 4. Experimental Design and Procedures

The experiment was programmed with z-Tree (Fischbacher [18]). The experimental sessions were run at the economics laboratory of the University of Erfurt (Germany). Each session comprised two independent observational groups with 12 subjects each. In each group, six males and six females participated. Subjects were invited with the recruitment software ORSEE (Greiner [19]) and were randomly matched to groups. Most participants were undergraduate students of social sciences. None of them had participated in a similar experiment before. In total, 168 subjects participated in 14 independent experimental groups.

As the participants arrived in the laboratory, they were seated in separate cabins and received a copy of the experimental instructions (see Appendix B). To ensure that common information on the experimental setup was given, the instructor read the instructions aloud. The same instructor was present at each session. After the instructions were read, the participants were allowed to ask clarifying questions on the rules of the game. On average, an experimental session took 2 h. Participants were paid privately. The conversion rate from tokens to cash was publicly announced in the instructions. The average pay was 21 Euro.

## 5. Results

In this section, we present our experimental results. On the aggregate level, our main variables of interest are: (i) the team choice, (ii) contributions to the public good (team project), (iii) the use of punishment options, and (iv) the resulting payoffs. We studied subjects' initial aggregate behavior, and how their decisions evolved over rounds. We were also interested in individual behavior, and examined the determinants of team choice, contributions, and punishment. We proceed in two steps. First, we look at behavior in the first period, where the decisions of other players cannot have influenced players' team and contribution choices. Second, we look at the dynamics of the behavior in the later periods.

*5.1. First Period Behavior*

5.1.1. Initial Team Choices

In the first period, women appeared to be rather indifferent to choosing between their two team options. In total, 56.0% of the women chose the WM-team, compared to 44.0% who chose the W-team ($p = 0.326$, binomial test).[5] A clear majority of men preferred the mixed-gender team in the first period: 70.2% of the men chose the WM-team, compared to 29.8% who chose the men-only M-team ($p < 0.01$, binomial test). While the second observation is in line with our Hypothesis 1b, the first one does not appear to be in line with Hypothesis 1a.

**Result 1:** *Initial Team Choice. In the first period, about half of the women choose the mixed-gender team while a clear majority of men chooses this team.*

Note that subjects did not have an incentive to choose the mixed-gender team if they assumed that others contribute equally in different teams. The reason is that the WM-team may potentially consist of

---

5   All tests in the result section report two-tailed significance levels. Because of the interdependence of the decisions, within and across teams, all following within and across team comparisons are tested with Wilcoxon matched-pairs test, using group (team) averages as independent observations. To save space, we abbreviate the Wilcoxon matched-pairs test to WMPT.

12 members and thereby be considerably larger than the same gender teams. As mentioned above in our design, the individual return from the public good decreased with the number of group members.

5.1.2. Initial Contributions, Punishment, and Pay

Figure 1 shows the average contributions, pay (profit)[6], and sent and received punishments in the first period, for both genders and all teams. Averaged over all teams, in period 1, women contributed 8.1 on average, which is significantly less than the men's contributions of 11.4 ($p = 0.026$, Wilcoxon matched-pairs test (WMPT)).[7] While women predominantly contributed 5 and 10, men predominantly contributed 15 or 20. The same is true if one focuses on the WM-team, where men contributed 10.9 on average while women contributed 7.6 on average, which is significantly less ($p = 0.027$, WMPT, see Figure 1, panel a). The contributions of the women do not significantly differ between the W-team and the WM-team ($p = 0.508$, WMPT). The same is true for men in the M-team and the WM-team ($p = 0.860$, WMPT). Thus, we did not find support, neither for Hypothesis 2a nor for Hypothesis 2b, i.e., that the contribution behavior of women and men differ between teams.

**Result 2:** *Initial contributions. In the first period, men contribute more than women do. Contributions of women do not differ between the two teams. The same is true for men.*

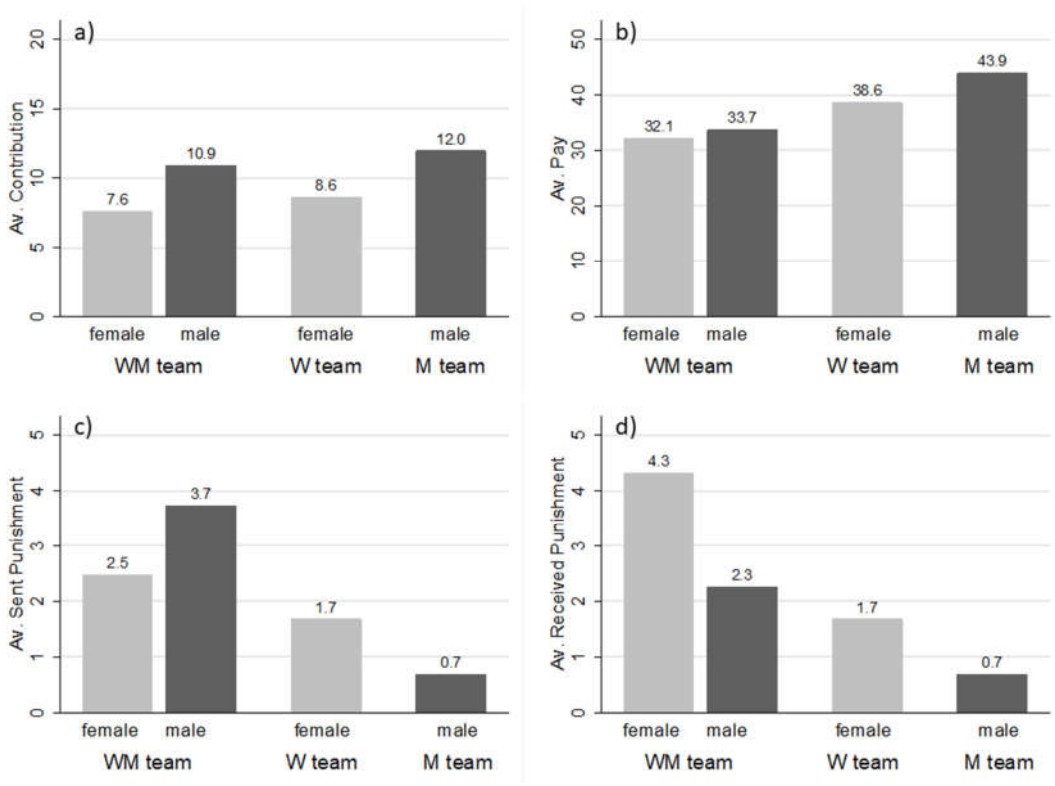

**Figure 1.** Average (**a**) contributions; (**b**) average pay; (**c**) average sent punishment points; and (**d**) average received punishment points; over all periods and teams, respectively.

As can be seen from panels (c) and (d) in Figure 1, the average punishment points sent and received per subject tend to be lower in the same-gender teams compared to the mixed gender teams.

---

6　Average pay is the net payoff after the punishment stage, including the lump-sum payment, also including possible costs for allocating punishment points and deductions due received punishment from others.

7　This finding is in contrast to Ortmann and Tichy [20], who report that women cooperate significantly more than men do in the first round of a repeated, binary choice prisoner's dilemma game in exogenously formed two-player teams. This difference disappears by the last period.

Indeed, women in the WM-team received significantly more punishment points compared to women in the W-team ($p = 0.047$, WMPT) and men in the WM-team sent weakly significantly more punishment points than men in the M-team ($p = 0.051$, WMPT). However, regression analyses revealed that by controlling for the contribution difference between the punisher and the punished individual, there are no differences for men's initial punishment behavior in the M-team and WM-team, while women punish more initially in the W-team than in the WM-team.[8]

**Result 3: *Initial punishment.*** *In the first period, women in the women-only team punish more than women in the mixed-gender team, even when controlling for the contribution difference between the punisher and the punished person.*

In the first period, women in the W-team earned 38.6, significantly less than men in the M-team with 43.9 ($p = 0.028$, WMPT, see Figure 1, panel b).[9] Averaged over all teams, in the first period women earned 35.4 on average while men earned 38.8 ($p = 0.096$, WMPT). Women in the WM-team earned significantly less than women in the W-team ($p = 0.023$, WMPT). The main reason is that the relatively low contributing women in the WM-team were punished more severely by the high-contributing men compared to the punishment women received from other women in the W-team. Also, men in the WM-team earned significantly less than men in the M-team ($p = 0.002$, WMPT). The main reason here is that fellow women in the WM-team tended to contribute less than fellow men in the M-team. A second reason is that men in the WM-team also tended to punish more than men in the M-team.

**Result 4: *Initial Pay.*** *In the first period, women earn more in the women-only team than in the mixed-gender team. The analogous result is also true for men. Women earn less in the women-only team than men do in the men-only team.*

*5.2. Behavior over Time*

5.2.1. Team Choices over Time

The bars in Figure 2 show the team choices of both genders. Panel (a) provides the average number of women (grey bars) and men (black bars) in the WM-team over all the periods. The grey bars in panel (b) show the corresponding average number of women in the W-team and the black bars in panel (c) show the corresponding average number of men in the M-team. As discussed above, and as can be seen from all three panels, many players chose the WM-team in period 1, but most of them were men. In periods two and three, both genders tended to move to the same-gender teams. Thus, in the third period, about 60% of each gender chose the exclusive-gender teams. This basically did not change for men in the remaining 27 periods. Women, however, increasingly moved to the mixed-gender team. In period 10, about 60% of the women chose the WM-team. This percentage tends to be stable for the next 15 periods after which more women started to move to the W-team, so that in period 30 about 50% of the women stayed in the W-team, and 50% of the women stayed in the WM-team. Overall, a woman switched on average 3.7 times from one team to the other while a man did so only 2.8 times ($p = 0.068$, WMPT). This observation is in line with Figure 2 which shows a relatively stable high number of men in the M-team.[10]

As one might expect, being punished increases the likelihood of switching to the other team. In a probit regression with a binary dependent variable CHANGE TEAMS, denoting a switch from one team to the other, the binary independent variable PUNISHED turns out to be highly significant (see regression (a) in Table 1). Women and men tended not to behave differently in this respect,

---

[8]   See Table A1 in the Appendix A.
[9]   This is in line with Rapoport and Chammah [21], who report that male-male interactions are more cooperative than female-female interactions.
[10]  Of all the possible groups, 8.9 percent consisted of only one subject in a given period. If there is only one subject in a group, then the single player's endowment is added automatically to their private account.

as suggested by the non-significant interaction variable PUNISHED X MAN. The regression (a) also confirms the impression mentioned above that men switch less often than women do, as indicated by the negatively significant indicator variable men.[11] The men's likelihood of switching to the other team tended to increase if the average contribution in the last period was higher in the team the subject did not select than in the team the subject actually selected (cf. the variable OTHER TEAM'S CONTRIBUTIONS HIGHER X MAN).

**Table 1.** Regression results on (a) team change behavior; (b) conditional cooperation; (c) received punishment; and (d) reaction to punishment.

| | (a)<br>DV = Change Teams | (b)<br>DV = Contribution | (c)<br>DV = Received Punishment | (d)<br>DV = Contribution Difference |
|---|---|---|---|---|
| punished | 1.030<br>(0.100) *** | | | |
| other teammates'<br>average contributions | | | −0.355<br>(0.077) *** | |
| other team's<br>contribution higher | 0.210<br>(0.135) | | | |
| man | −0.264<br>(0.124) ** | 1.102<br>(2.411) | | −0.243<br>(0.347) |
| punished x man | −0.102<br>(0.085) | | | |
| other team's average<br>contributions higher x man | 0.341<br>(0.204) * | | | |
| others' average<br>contributions in the<br>own team | | 1.272<br>(0.108) *** | | |
| others' average<br>contributions in the own<br>team x man | | −0.026<br>(0.137) | | |
| neg. contribution diff.<br>between the subject and<br>the teammates | | | 1.257<br>(0.098) *** | |
| pos. contribution diff.<br>between the subject and<br>the teammates | | | −0.116<br>(0.897) | |
| received punishment<br>(previous period) | | | | 0.321<br>(0.054) *** |
| received punishment<br>(previous period) x man | | | | 0.104<br>(0.146) |
| constant | −1.501<br>(0.136) *** | −0.592<br>(1.125) | 1.252<br>(0.931) | 0.576<br>(0.189) *** |
| *N* | 4763 | 4749 | 4928 | 931 |
| Std. err. adjusted for 14 groups | Yes | Yes | Yes | Yes |

Panels employ the following models: (a) Random effects probit model with change teams 'yes' = 1 or 'no' = 0, (b) Tobit model with contribution as the dependent variable (DV), (c) Tobit model with received punishment as the DV, and (d) Tobit model with CONTRIBUTION DIFF. between period t and period t − 1. All models are clustered by independent observation groups (including six women and six men each, respectively). Models (a)–(c) consider only those cases in which more than one single person was in a team. Contribution and punishment possibilities were available only in such cases. Model (d) refers to "contribution reactions" only in cases where the player received a non-zero punishment in the previous period. * $p < 0.1$; ** $p < 0.05$; *** $p < 0.01$.

If we focus on the WM-team, women were significantly more likely to leave when being punished than when they were not punished (37.2% vs. 17.0%, $p = 0.007$, WMPT). Men also left the WM-team when being punished (32.6%), but not significantly more often than when they were not punished (27.8%, $p = 0.340$, WMPT). As suggested by the regression result above, one reason for this observation might be that men predominantly leave the WM-team for the higher contributions in the M-team. Thus, team choice behavior over time does not support our Hypothesis 1b.

---

[11]　Halladay [22] finds women increase performance when experiencing negative emotions, while male performance remains unaffected.

**Result 5:** *Team choice over time. Overall, women quite frequently switch between teams and exhibit a slight preference for the mixed-gender team. Men switch less often and tend to prefer the men-only team.*

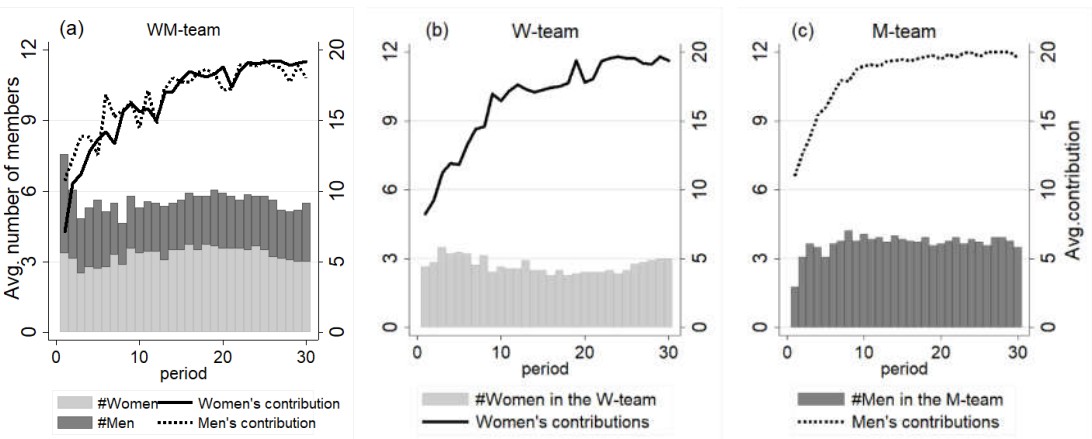

**Figure 2.** Average numbers (bars) and average contributions (lines) of women and men in each period. Bars in panel (**a**) provide the average number of women (grey bars) and men (black bars) in the WM-team over periods. The grey bars in panel (**b**) show the corresponding average number of women in the W-team and the black bars in panel (**c**) show the corresponding average number of men in the M-team.

5.2.2. Contributions over Time

As in period 1, and also averaged over all periods and teams, women contributed significantly less than men do (women contributed 16.9 on average and men contributed 18.1 on average; $p = 0.022$, WMPT).[12] The difference in contributions between genders was mainly due to contribution differences in the same-gender teams. Averaged over all periods, women contributed 17.0 in W-teams, while men's average contributions in M-teams (18.5) were significantly higher ($p = 0.036$, WMPT). The difference in contributions between women and men was particularly pronounced during the first half of the periods 1–15 (on average, women contributed 14.7, and men 16.8; $p = 0.013$). In the second half, the difference in contributions was no longer significant (women contributed 19.0 and men contributed 19.4; $p = 0.572$, WMPT). The development of contribution differences is shown in more detail in Figure 3, panel (a).

Thus, contributions of both genders tended to converge towards each other over as the periods evolved. It is also true that on average, women and men exhibited a similar extent of conditional cooperation. The regression (b) in Table 1 explaining the contributions in period t shows that they are highly correlated with the respective average contributions of others in the same team in period $t - 1$ (cf. the variable OTHERS' (PREVIOUS) CONTRIBUTIONS IN THE OWN TEAM. This suggests that participants were conditionally cooperative. In fact, both genders seemed to be conditionally cooperative to a similar extent since the interaction variable OTHERS' (PREVIOUS) CONTRIBUTIONS IN THE OWN TEAM X MAN turns out not to be significant. Again, also over time, we did not find differences in contributions between teams, neither for women nor for men, which is not in line with our Hypotheses 2a and 2b.

**Result 6:** *Contributions over time. Averaged over all periods, men contribute more than women do, but the difference in contributions disappears over periods. Women and men are quite similar in the extent of their conditionally cooperative behavior.*

---

[12] For a visual comparison of average contributions, pay, sent punishment points, and received punishment points, averaged over all periods for each of the teams, respectively, see Figure A1 in the Appendix A.

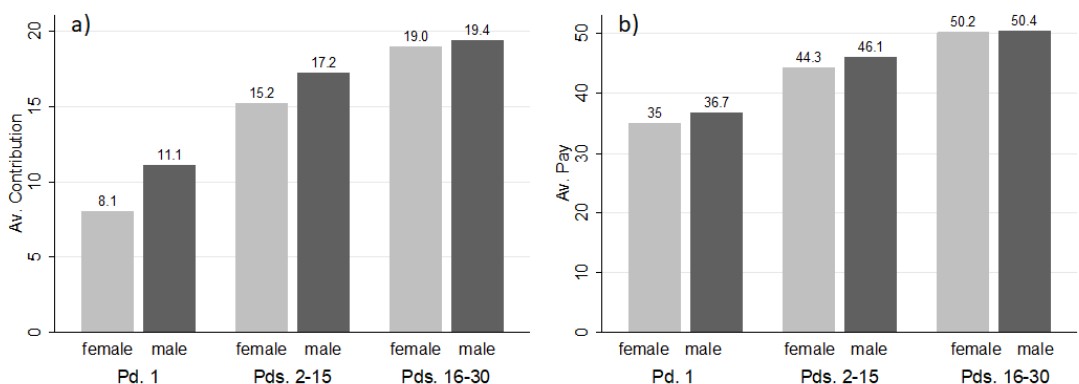

**Figure 3.** Average contributions (panel **a**) and pay (panel **b**) of women (grey bars) and men (black bars) over different periods. The first pair of bars shows the averages in the first period. The second pair of bars shows the averages over periods 2–15, and the third pair of bars shows the averages over periods 16–30.

### 5.2.3. Punishment Behavior over Time

Overall, women on average sent 0.66 punishment points while men sent 0.80 points (per period). Women received 0.82 punishment points, and men received 0.64 punishment points (per period). Both differences are not significant ($p = 0.397$, and $p = 0.158$, respectively, WMPT). Women sent and received only slightly more punishment points (on average 0.6) in W-teams, compared to men in M-teams (on average 0.5). There was, however, a difference in WM-teams, where men sent on average significantly more punishment points than women (women sent 0.7 on average, and men sent 1.2 on average; $p = 0.004$, WMPT).

The Tobit regression model shown in panel (c) in Table 1 explains the number of received punishment points as the dependent variable, with the independent variables OTHER TEAMMATES' AVERAGE CONTRIBUTIONS, the absolute value of the NEGATIVE CONTRIBUTION DIFFERENCE BETWEEN THE PUNISHED & THE TEAMMATES, and the POSITIVE CONTRIBUTION DIFFERENCE BETWEEN THE PUNISHED & THE TEAMMATES, as also used by Fehr and Gächter [23] in a similar regression (see Table 5 on page 991). The corresponding coefficients indicate a positive correlation between the negative deviation from fellow teammates' average contributions and received punishment, while there is no such relationship for positive deviations. In other words, while we observed that "social punishment" depended positively on the contribution difference, we did not observe such a relationship for "antisocial punishment" (which is in line with Fehr and Gächter's analysis). Table A2 in the Appendix A provides the estimates of model (c) for males and females, showing that the results discussed above also hold for both genders separately.

The received punishment points in WM-teams were higher for women (on average 1.0) compared to men (on average 0.8). However, the difference is only weakly significant ($p = 0.084$, WMPT).[13] Panel (d) in Table 1 reports the results of a Tobit regression model explaining the CONTRIBUTION DIFFERENCE between period t and t − 1 using the independent variable RECEIVED PUNISHMENT (PREVIOUS PERIOD), the amount of punishment received in period t − 1. Being punished seems to lead to an increase in contributions. This effect did not seem to be different between genders, as the non-significant coefficient of the interaction variable RECEIVED PUNISHMENT (PREVIOUS PERIOD) X MAN indicates.

---

13   If we control for the contribution difference between the punisher and the receiver, neither men nor women discriminate between genders when punishing. In other words, women do not punish other women more than men punish women. Men do not punish other men not more severely than women punish men.

**Result 7:** *Punishment over time. Averaged over all periods and teams, sent punishment and received punishment tend to be similar for women and men. Also, the reaction to punishment in terms of contribution change seems not to be different between women and men. Only in the mixed-gender teams, men tend to exert more punishment than women do.*

5.2.4. Pay

As in the first period, women's average pay over all 30 periods was significantly lower than that of men (women earned 46.9 on average and men earned 47.9; $p = 0.026$, WMPT). This was primarily driven by the (weakly significant) lower pay of women in the W-teams compared to men's pay in the M-teams (on average, women earn 47.5 and men earn 49.1; $p = 0.084$, WMPT). Women in the WM-teams and the W-teams earned non-significantly different amounts ($p = 0.925$, WMPT). The same is true when one compares pay for men in the WM-teams and M-teams ($p = 0.245$, WMPT). The difference in pay between genders comes into being in periods 1–15. The average pay of women in these periods over all teams was significantly lower than the average pay of men (women earned 43.6 and men earned 45.4; $p = 0.019$, WMPT). This is not the case in the second half of the periods. There was no significant difference in pay averaged over periods 16–30 (women earn 50.2 and men earn 50.4; $p = 0.594$, WMPT).

**Result 8:** *Pay over Time: Over all periods, women earn less than men. This is particularly true when comparing the same-gender teams. The difference in pay, however, disappears over time.*

**6. Conclusions**

We studied gender diversity and team performance in endogenously formed teams in a repeated team game. In each repetition, male and female participants could freely choose whether to perform a task in a team solely with members of the same (own) gender (women-only and men-only), or whether to perform the task in a mixed-gender team. We found an endogenously occurring "gender pay gap", driven by gender differences in the initial behavior. Independent of the team choice, men contributed significantly more to the team project than women.[14] These initial differences seemed to induce men to predominantly choose the more successful men-only team in the further periods, resulting in significantly higher profits for men than for women. It took until the second half of the thirty repetitions for women to "recover" from the disadvantage of the low initial contributions and for the "gender profit gap" to close.

We provide evidence that these results cannot be explained by lower cooperativeness of women per se: men and women did not exhibit different degrees of conditional cooperation in response to the groups' past contribution level, and they also did not exhibit differences in punishment behavior. In the first period, however, when a group's cooperativeness is unknown, cooperation is a matter of risk and trust. In their review paper, Croson and Gneezy [9] discuss gender differences in trust games and conclude that "women trust less than or the same as men in these settings. However, women's trust levels are more context-sensitive than those of men" (p. 460), in line with Gilligan [27]. Simpson and Van Vugt [28] describe two motivations for non-cooperation: greed and fear. Greed corresponds to the temptation to free-ride on others' cooperation, while fear refers to the prospect that one's cooperation may be exploited. They argue that from an evolutionary point, these motivations affect women and men differently. From an evolutionary perspective, women avoid taking too many risks to avoid being exploited in social interactions, while men have evolved a "high-risk-high-stakes" strategy. As a consequence, they argue that women's non-cooperation is mainly due to the fear of being exploited by others, while men's non-cooperation is motivated by greed. Our observation that

---

[14]    The question of whether women or men contribute more in a public goods game provided inconclusive answers. While some studies find—as we do—men to be more cooperative than women (e.g., Brown-Kruse and Hummels [16]) others find the opposite (e.g., Charness and Villeval [24]) or report no differences (e.g., Bolton and Katok [25], Eckel and Grossman [26]). See Croson and Gneezy [9] for an extensive discussion on this issue.

men's main motivation for switching the team seems to be a because of higher profit in the other team, nicely fits into this picture. Even more importantly, this evolutionary perspective also fits the observed low cooperativeness of women in the first period (out of fear of being exploited) and the lack of difference in conditional cooperation between women and men in later rounds when experience allows better calibration of the risk of being exploited.

We are aware that our study also comes with limitations. Experimental studies in a laboratory with mainly students as participants raise questions of external validity. Thus, complementary evidence from the field would be highly valuable. Our teams are rather ad hoc and did not interact for long. Team members are completely anonymous and cannot be identified as single individuals. This would rarely be the case in teams outside the laboratory. Also, the punishment technology is quite abstract and focusses on monetary consequences. Punishment for norm enforcement often is more subtle, indirect, and often without money involved, e.g., mobbing or social isolation at the workplace. In our study we use gender as the only differentiating characteristic among the two groups. Thus, it is unclear whether, for example, a minimal group paradigm primed to induce two groups without referring to gender would already yield similar findings. Nonetheless, the controlled and abstract environments can yield important clues about structural regularities.

Our results point to the crucial importance of identifying contexts that encourage women to initially be as cooperative as men since low initial contributions have detrimental effects for women's profits, which only improve slowly. Our data suggest that creating a "reserved space for women", i.e., creating a context that ensures that women interact with other women only, is not sufficient. It seems that more rigorous measures have to be taken to initially foster women's trust and remove the initial gender contribution difference. In this respect, our results point to a new direction for future avenues in gender research.

**Author Contributions:** Conceptualization, Ö.G., B.I. and B.R.; methodology, Ö.G., B.I. and B.R.; software, Ö.G.; validation, Ö.G., B.I. and B.R.; formal analysis, Ö.G., B.I. and B.R.; investigation, Ö.G., B.I. and B.R.; data curation, Ö.G., B.I. and B.R.; writing—original draft preparation, Ö.G., B.I. and B.R.; writing—review and editing, Ö.G., B.I. and B.R.; funding acquisition, Ö.G., B.I. and B.R.

**Funding:** This research was funded by the German Science Foundation, grant number DFG-RO 3071/1-2.

**Acknowledgments:** We thank the participants of the 2012 European Meeting of the Economic Science Association in Cologne, and the participants of the 2017 Annual Conference of the Verein für Socialpolitik in Vienna for helpful comments.

**Conflicts of Interest:** The authors declare no conflict of interest. The funders had no role in the design of the study; in the collection, analyses, or interpretation of data; in the writing of the manuscript, or in the decision to publish the results.

## Appendix A

**Table A1.** Regression results on sent punishment for (a) men and (b) women.

| DV = Sent Punishment | (a) Tobit Regression Punisher = Man | (b) Tobit Regression Punisher = Woman |
|---|---|---|
| contribution difference btw. punisher & the punished | 0.593 (0.215) *** | −0.131 (0.111) |
| wm-team (dummy) | 0.331 (1.590) | −1.954 (0.828) ** |
| contribution difference btw. punisher and & the punished x wm-team | −0.239 (0.184) | 0.283 (0.137) ** |
| constant | −4.324 (2.277) * | 1.082 (0.698) |
| *N* | 228 | 162 |
| Std. err. adjusted for clusters | Yes | Yes |

Panels employ the following models: Tobit model with sent punishment as the dependent variable (DV) for (a) men as senders and (b) women as senders; all models are clustered by independent observation groups (6 women and 6 men, respectively). * $p < 0.1$; ** $p < 0.05$; *** $p < 0.01$.

**Table A2.** Regression results on received punishment by (a) men and (b) women.

| DV = Received Punishment | (a) Tobit Regression Punished = Man | (b) Tobit Regression Punished = Woman |
|---|---|---|
| other teammates' average contributions | −0.438 (0.083) *** | −0.301 (0.073) *** |
| neg. contribution diff. between the subject and the teammates | 1.227 (0.106) *** | 1.269 (0.134) *** |
| pos. contribution diff. between the subject and the teammates | −0.147 (0.136) | −0.131 (0.132) |
| constant | 3.127 (1.245) ** | 0.158 (0.694) |
| N | 2029 | 1912 |
| Std. err. adjusted for clusters | Yes | Yes |

Panels employ the following models: Tobit model with received punishment as the dependent variable (DV) for (a) men as senders and (b) women as senders; all models are clustered by independent observation groups (6 women and 6 men, respectively). * $p < 0.1$; ** $p < 0.05$; *** $p < 0.01$.

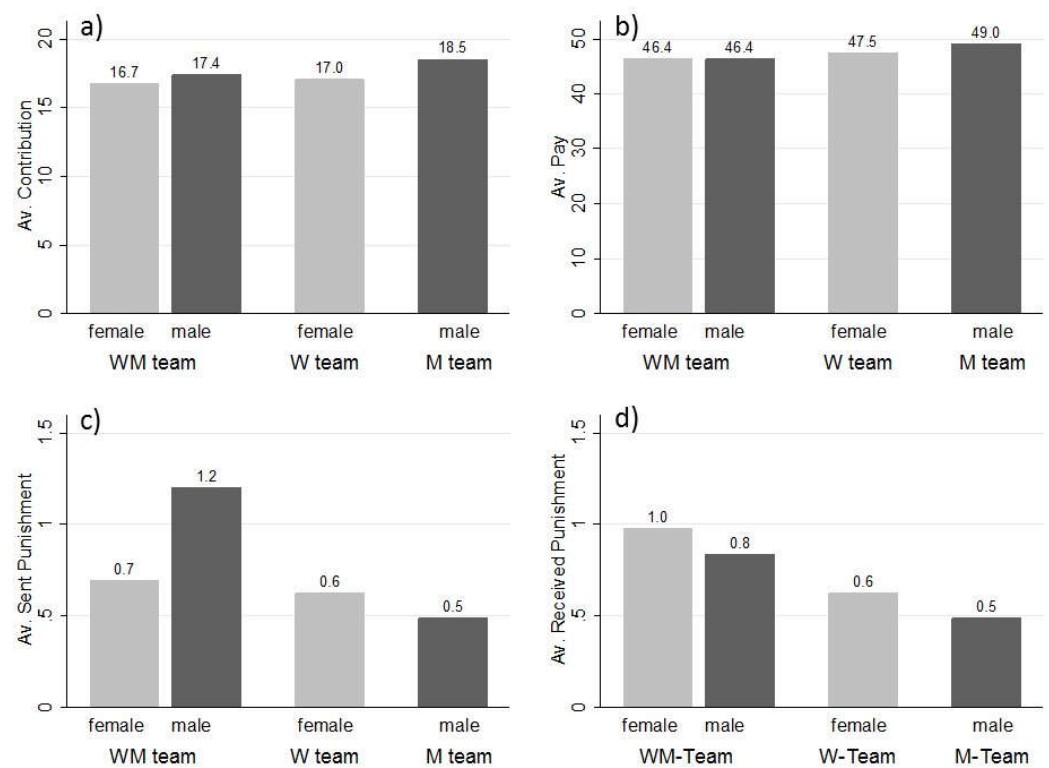

**Figure A1.** Average (**a**) contributions; (**b**) average pay; (**c**) average sent punishment points; and (**d**) average received punishment points; over all periods and teams, respectively.

## Appendix B

### Instructions for the Experiment

Original instructions were provided in German. They are available from the authors on request.

**General Information:** At the beginning of the experiment, you will be randomly assigned to one of two subpopulations, each consisting of 12 participants. During the whole experiment, you will interact only with the members of your subpopulation. To each of the subpopulations, 6 men and 6 women will be assigned.

**The course of Action:** The experiment consists of 30 rounds. Each round consists of two stages. In Stage 1, the group choice and the decision regarding the contribution to the project take place. In Stage 2, participants may influence the earnings of the other group members.

**Stage 1**

**(i) The Group Choice:** In Stage 1, each participant decides which group she/he wants to join. There are three different groups:

- G-Group: Women, as well as men, can enter this group.
- F-Group: Only women can enter this group.
- M-Group: Only men can enter this group.

Each participant can choose between two of these three groups. Men can choose between the M-Group and the G-Group. Women choose between the F-Group and the G-Group.

After the group choice has been completed, you will learn how many members each group has. For the G-Group, you will also learn how many women and men are in that group.

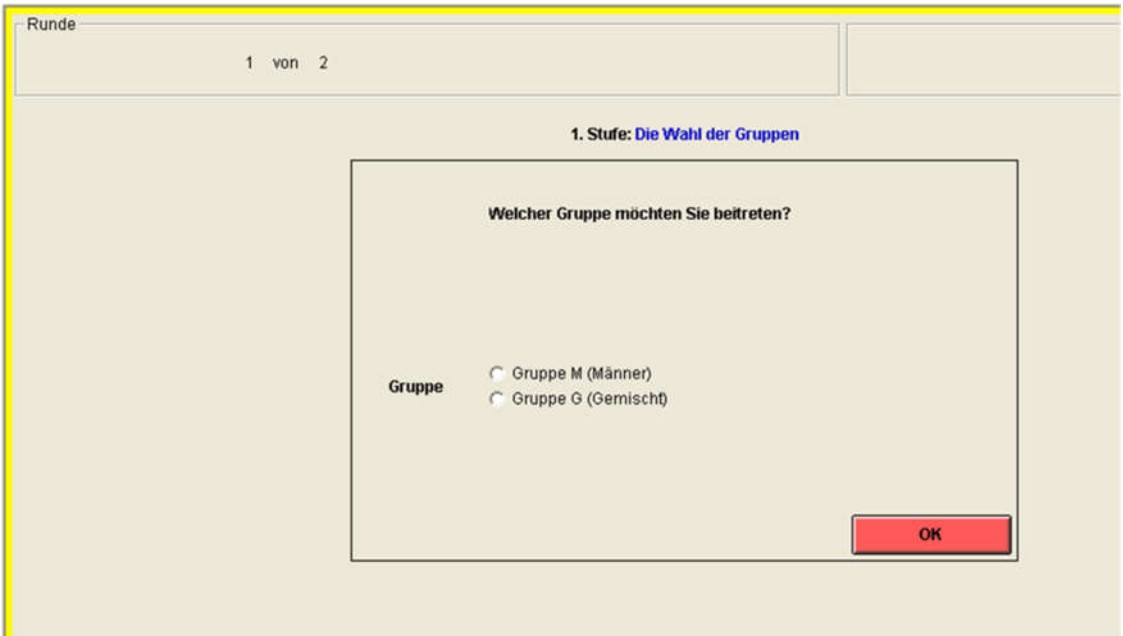

**(ii) Contributing to the Project:** In Stage 1 of each round, each group member is endowed with 20 tokens. You have to decide how many of the 20 tokens you are going to contribute to the project. The remaining tokens will be kept by you.

**Your Earnings from the project:**

=1.6 × Sum of the Contributions of all Group Members/Number of Group Members

For each group member, the earnings from the project are calculated according to this formula. Please note: Each group member receives the same earnings from the project, i.e., each group member benefits from **all** contributions to the project.

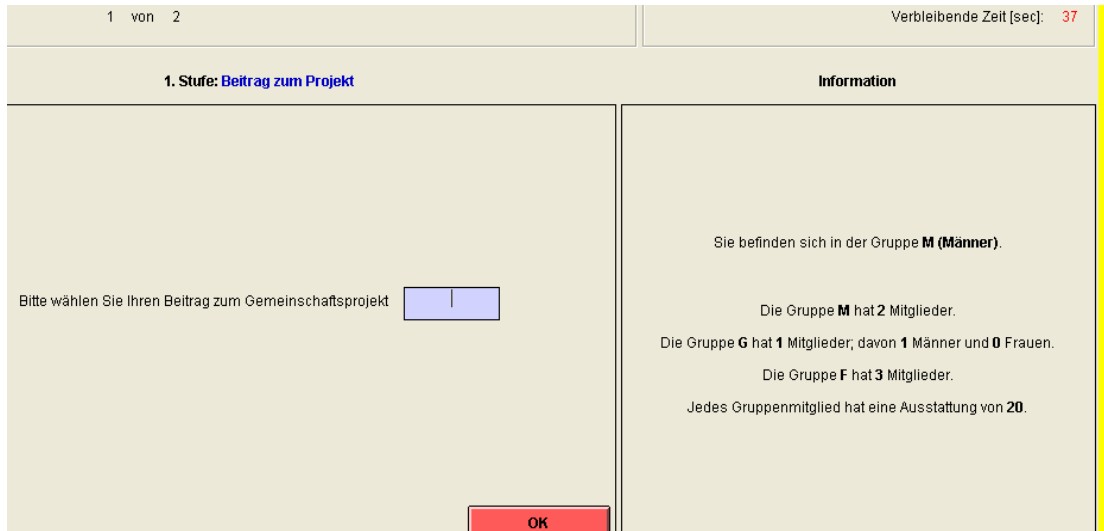

**Calculation of your payoff in Stage 1:** Your payoff in Stage 1 consists of two components:

- **tokens you have kept** = endowment − your contribution to the project
- **earnings from the project** = 1.6 × sum of the contributions of all group members/number of group members

| | Your Contribution to the Project | The Sum of Others' Contributions in Your Group | Number of Team Members in Your Group | Your Earnings from the Project | Your Earnings from Stage 1 |
|---|---|---|---|---|---|
| **Example 1** | 7 | 45 | 5 | 45 × 1.6/5 = 14.4 | 20 − 7 + 14.4 = 27.4 |
| **Example 2** | 1 | 33 | 7 | 33 × 1.6/7 = 7.5 | 20 − 1 + 7.5 = 26.5 |
| **Example 3** | 16 | 67 | 4 | 67 × 1.6/ 4= 26.8 | 20 − 16 + 26.8 = 30.8 |

## Stage 2

**Assignment of Tokens:** In Stage 2, it will be displayed how much each group member has contributed to the project. In the G-Group, you will also learn the gender of the respective member.

By assigning tokens, you can reduce the payoff of a group member or keep it unchanged.

In each round, each participant receives an additional 20 tokens in Stage 2. You have to decide how many of the 20 tokens you are going to assign to other group members. The remaining tokens are kept by you. You can check the costs of your token assignment by pressing the button Calculation of Tokens.

- Each negative token you assign to a group member **reduces her payoff by 3 tokens**.
- If you assign 0 tokens to a group member, her/his **payoff won't change**.

**Please note: Before each round, a display order will randomly be determined.** Thus, it is not possible to identify any group member by her position on the displayed list throughout different rounds.

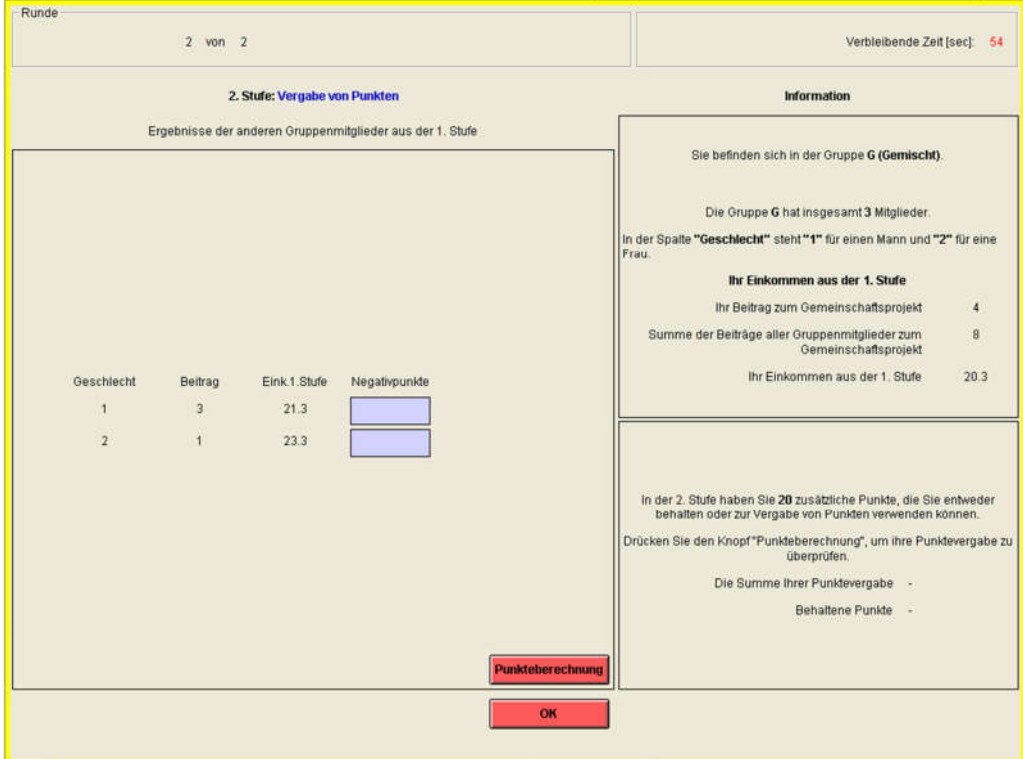

**Calculation of your payoff in Stage 2:** Your payoff in Stage 2 consists of two components:

- **tokens you have kept** = 20 – sum of the tokens that you have assigned to the other group members
- **less the threefold number of negative tokens** you obtained from other group members

**Box A1**

> Thus, **your payoff in Stage 2** amounts to:
> 20 – sum of the tokens that you assigned to other group members – 3 × (the number of tokens you obtained from other group members)

**Calculation of your round payoff:** Your round payoff is composed of

Your payoff from Stage 1 20 – your contribution to the project + 1.6 × sum of the contributions of all group members/number of group members
+ Your payoff from Stage 2 20 – sum of the tokens you have assigned to other group members – 3 × (number of tokens you obtained from other group members)
= Your round payoff

**Special case, a single group member:** If it happens that you are the only member in your group, you will receive 20 tokens in Stage 1 and 20 tokens in Stage 2, i.e., your round payoff will amount to 40. You do not have to take any action on either Stage 1 or Stage 2.

**Information at the end of the round:** At the end of the round, a detailed overview will inform you of the results obtained in all groups. For each group member, you will learn the following about that person: His/her contribution to the project, payoff from Stage 1, assigned tokens (if applicable), received tokens (if applicable), payoff from Stage 2, round payoff, and for the G-Group you will learn the gender of the respective group member.

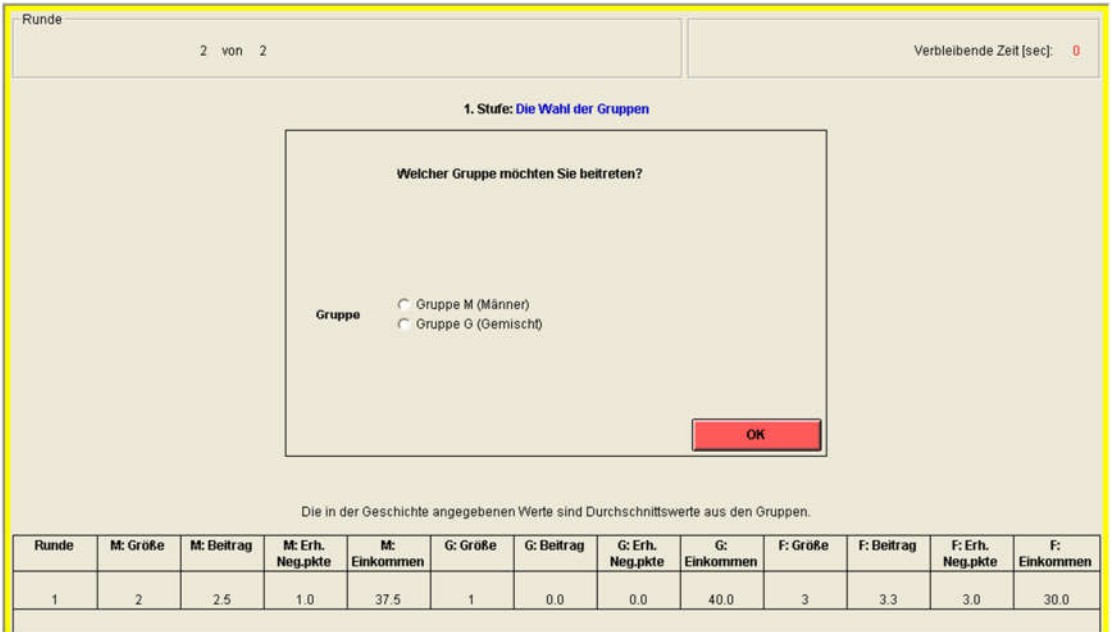

| Rundenergebnisse der Mitglieder der Gruppe M | | |
|---|---|---|
| Beitrag | 1 | 4 |
| Eink.1.Stufe | 23.0 | 20.0 |
| Verg.Pkte | -1 | -1 |
| Erh.Neg.pkte | -1 | -1 |
| Rundeneink. | 39.0 | 36.0 |

| Rundenergebnisse der Mitglieder der Gruppe G (Anmerkung zur Zeile Geschlecht: 1=Mann, 2=Frau) | |
|---|---|
| Geschlecht | 1 |
| Beitrag | 0 |
| Eink.1.Stufe | 20.0 |
| Verg.Pkte | 0 |
| Erh.Neg.pkte | 0 |
| Rundeneink. | 40.0 |

| Ihr Rundenergebnis (F) | | Rundenergebnisse der anderen Mitglieder der Gruppe F | |
|---|---|---|---|
| Beitrag | 5 | 3 | 2 |
| Eink. 1. Stufe | 20.3 | 22.3 | 23.3 |
| Verg.Pkte | -3 | -3 | -3 |
| Erh.Neg.pkte | -3 | -2 | -4 |
| Rundeneink. | 28.3 | 33.3 | 28.3 |

**History:** Starting from the second round, at the beginning of a new round, you will be informed about the overview of the average results (as above) of all previous rounds.

**Total Payoff:** The total payoff from the experiment is composed of the starting capital of 400 tokens plus the sum of round payoffs from all 30 rounds. At the end of the experiment, your total payoff will be converted into Euro with an exchange rate of 1.15 € per 100 tokens.

**Please notice:** No communication is allowed during the whole experiment. If you have a question, please raise your hand outside of the cabin. All decisions will be made anonymously, i.e., no other participant will be informed about the identity of anyone who has made a particular decision. Payment will be anonymous, too, i.e., no participant will find out what the payoff of another participant was.

We wish you every success!

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
