# Peer review of "Endogenously Emerging Gender Pay Gap in an Experimental Teamwork Setting"

_games, doi:10.3390/g9040098_

Round 1
Reviewer 1 Report
Short summary:
The paper describes a laboratory experiment designed to explore endogenous team formation in a repeated public goods game with peer punishment when gender is salient. In each round, participants first decide whether to join a team of their own gender, or a mixed-gender team. The authors find an initially higher cooperation and payoff in men compared to woman, yet the difference vanish over the course of the experiment. Similarly, initial differences in punishment are reported, but they do not persist when taking into account all periods.
Comments and suggestions:
It would be helpful to better explain and justify the non-parametric testing strategy used throughout the paper. In particular, the Mann-Whitney test (Page 5, line 180) isn’t usually used for categorical data. Additionally, what is treated as independent observation for the comparison of contributions (page 9, line 279) and punishment (page 9, line 306).
Page 5, line 193: How is “average pay” calculated? Specifically, is it the payoff after the punishment stage, for which the cost of punishing others and the received punishment is already deduced? Does it include the 20 tokens lump-sum payment?
Page 5, line 205: “… average punishment points sent and received per subject tend to be lower in the same-gender teams, compared to the mixed gender teams”. The difference in the average punishment points sent in the M-team vs. the WM-team is very interesting. However, the average contributions in the M-team is also higher compared to the WM-team. I think that, instead of a simple comparison of means, a regression analysis should be used which controls for the deviation of the punished person from the punisher’s contribution (see for example Table 5 in Fehr and Gaechter 2000 AER).
Page 6, line 210: Result 3 seems to be summarising contribution behaviour, not initial punishment.
Page 6, Figure 1: Please label the y-axes.
Page 7, Figure 2a: It would be helpful to include the second y-axis for the reader to be able to assess the contribution levels.
Page 7, line 250: “The men’s likelihood of switching to the other team tends to increase if the average contribution in the last period was higher in the team the subject did not select.” Did participants learn the average earning in the other teams?
Page 8, Table 1: Where observations excluded from the regression analysis? For regression model (b), I would expect 168 * 29 = 4872 observations. Please provide a justification in case observations were excluded. The same holds for the other regression models. Especially for regression model (c), I expected a larger number of observations because each participant takes a punishment decision regarding each other group member.
Page 10, line 312: I think it would make sense to split the variable delta_to_other_contr into positive and negative deviations. This would allow to test whether “perverse” / “antisocial” punishment exists, that is, the punishment of high contributors.
Reviewer 2 Report
REVISION
Comments to the Author(s)
The paper is entitled “Endogenously Emerging Gender Pay Gap in an Experimental Teamwork Setting”. This manuscript his study examine the relationship between gender diversity and performance in endogenously formed teams in a repeated 10 teamwork setting. Thus, I consider that the following considerations should be taken into account.
This research addresses an interesting and current topic and thanks for your submission to this journal. I wish the following points or suggestions would be helpful for you to further improve this manuscript.
(1) The abstract does not clearly describe the paper contents and hypotheses. The author(s) should be more specific in explaining the main aim of the manuscript as well as the most relevant implication of the research conducted. Moreover, abstaract does not include the sample used in this study.
(2) Introduction section is not clear, author(s) explain more information about gender diversity in positions on responsibility and their consequences in the performance. However, this section should include the following suggestions:
a. Authors do not present the main contributions of this study.
b. The last paragraph of this section should include the explanation of following points of this study. For example:
“The article is organised as follows. In the next section, we describe the institutional setting and in section three, we review the main theoretical ideas. Section four provides the hypotheses and section five describes the empirical design. Section six contains the results and finally, in section seven, we summarise and conclude.”
c. In the page 2, second paragraph, author(s) explain the main goal and the results obtained. However, they do not give information about the sample, duration of the study, experiments, among other questions.
Also, I think the introduction would greatly benefit from writing a clear research question that motivates the paper. An additional suggestion that can help better frame the introduction is to consider presenting the role of female in different fields as education, companies, game, among others (which I think can be a potential contribution of the manuscript).
Moreover, it is necessary to include the section “Institutional background” to explain the how the German legal system could influence on their manuscript, since authors should give information about gender diversity regulations of the country where they have done the experiment. This manuscript does not provide information about laws and regulations focused on gender diversity in positions of responsibility and their effect in the decision making process in Germany.
(3) Experimental design section should be the third section later theoretical background and hypotheses. This section is clear and well written.
(4) Theoretical considerations, author(s) do not include theories which support the main goal of this study, and the hypotheses proposed in page 4 are not supported with theories and past research. This study is very relevant for the previous literature although there is scant past research which examines this issue. For this reason, this study should give significant results and contributions.
(5) Experimental procedures should be a part of Empirical Design section. Author(s) should improve the following points:
a. Author(s) should explain the initial and final sample of participants of this experiment. So, I recommend them to include a “sample table”.
b. This sections are well written and presented. I also recommend the author(s) elaborating a table in which it is described how the variables are labelled, how they are measured or a short description and how they are expected to affect the dependent variables.
(6) Results and Conclusion section. Author(s) offer an extensive section clear and concise. However, the results obtained are not justified with theories and past research. In the conclusions section, author(s) should improve the implications paragraph. Finally, the authors should include the limitations of this study.
(7) References:
a. I recommend include "Games” papers since only includes one.
b. The references are not adapted to this journal.
-The tables presented are easy to understand. To facilitate reading, I would recommend the author to include the legend with the variables used.
I hope that my comments will help you in your revision of the manuscript. I wish you all the best during the revision process.
Round 2
Reviewer 2 Report
Thanks for allowing me to read this very interesting piece of research. I think that authors have done a good revision of the paper. I consider this very well written piece and I salute you for this. It meets all requirements for publication in the Games.